# Electrocatalytic Oxidation of Nitrophenols via Ag Nanoparticles Supported on Citric-Acid-Modified Polyaniline

**Milad Khani [1], Ramaswami Sammynaiken [2,*] and Lee D. Wilson [1,*]**

[1] Department of Chemistry, University of Saskatchewan, 110 Science Place, Saskatoon, SK S7N 5C9, Canada
[2] Saskatchewan Structural Sciences Centre, 110 Science Place, Saskatoon, SK S7N 5C9, Canada
* Correspondence: r.sammynaiken@usask.ca (R.S.); lee.wilson@usask.ca (L.D.W.); Tel.: +1-306-966-4733 (R.S.); +1-306-966-2961 (L.D.W.)

**Abstract:** Citric-acid-modified polyaniline (P-CA) and P-CA modified with Ag nanoparticles (Ag@P-CA) were prepared via an in situ reduction method. The physicochemical properties of P-CA and Ag@P-CA were compared to unmodified polyaniline (PANI) and PANI-modified Ag nanoparticles (Ag@PANI). Ag@P-CA had a lower content of aniline oligomers compared to Ag@PANI. P-CA and Ag@P-CA had a greater monolayer adsorption capacity for 2-nitrophenol and lower binding affinity as compared to PANI and Ag@PANI materials. X-ray photoelectron spectroscopy and cyclic voltammetry characterization provided reason and evidence for the higher conductivity of citric-acid-modified materials (P-CA and Ag@P-CA versus PANI and Ag@PANI). These results showed the potential utility for the optimization of adsorption/desorption and electron transfer steps during the electrochemical oxidation of nitrophenols. The oxidation process employs Ag@P-CA as the electrocatalyst by modifying polyaniline with Ag nanoparticles and citric acid, which was successfully employed to oxidize 2-nitrophenol and 4-nitrophenol with comparable selectivity and sensitivity to their relative concentrations. This work is envisaged to contribute significantly to the selective conversion of nitrophenols and electrocatalytic remediation of such waterborne contaminants.

**Keywords:** electrocatalyst; polyaniline; silver nanoparticle; citric acid; nitrophenols; cyclic voltammetry; oxidation process

## 1. Introduction

Composites are an interesting branch of materials science since such materials combine the characteristics of their constituents or display entirely new functional properties that widen their field of application [1]. Polyaniline (PANI) is one of the most widely used electrically conductive polymers due to its facile synthesis, low cost, high yield, and high electrical conductivity [2,3]. The distinct electrical and electrochemical characteristics of PANI have led to its utility in various applications, including light-emitting diodes, plastic batteries, solid-state sensors, harmonic generators, energy storage devices, and anti-static and anti-corrosion coatings [4–6]. PANI comes in variable oxidized and protonated states with varying electrical conductivity, which can be tailored for various applications (Figure 1). The most stable form of PANI is the half-oxidized (emeraldine) form, which is primarily utilized in PANI-based electrocatalyst materials due to its favorable electrical conductivity [7]. Both leucoemeraldine and pernigraniline forms of PANI are unstable and only exist under certain conditions.

PANI's electrical conductivity and structural properties are influenced by the synthetic method employed. In addition, surface area and porosity [8], different dopant anions (e.g., chloride, sulphate, and polystyrene sulfonate) [9], and water vapor [9] have an influence on the electrical conductivity of PANI. Among various doping acids, organic acids (OAs) impart interesting properties to PANI, according to changes in the solubility and conductivity of doped PANI [10]. This effect of the dopant can be explained by

three factors: (i) OAs may function as a surfactant that serves to increase the solubility of PANI; (ii) the functional groups of OAs may enhance polymer solubility; (iii) OAs induce greater charge delocalization and conductivity of PANI [11]. OAs can also be considered as PANI-template structure directors due to RCOOH and/or ROH functional groups since OAs can result in a specific thickness and length of PANI, which influence the resulting physicochemical and structural properties [12]. The thickness of PANI nanostructures relates to the role of hydrogen bonding [12], which increases with a greater number of COOH groups for the OAs. OAs can induce aggregation of PANI nanostructures, which increase the complexity of the PANI surface sites. Among the various OAs, citric acid (CA) is a low-cost and safe synthon with three COOH groups and one OH group, which can be considered as a good candidate for modification of PANI [13].

**Figure 1.** Molecular structure of PANI in variable oxidation states: leucoemeraldine (LE), emeraldine base (EB), and pernigraniline (PG) [14].

Ag nanoparticles (NPs) with high electrical conductivity can be used as a transparent electrode material and catalyst for electrochemical reactions [15]. Incorporating Ag NPs onto various substrates can effectively minimize the aggregation of NPs and enhance their surface area and catalytic activity [16,17] for Ag-based catalysts. These unique materials are more efficient and environmentally friendly due to the stabilization of Ag NPs, which inhibits the loss of this valuable catalytic metal [18]. Therefore, PANI-citric acid (P-CA) can be utilized as a support for Ag NPs to reduce or oxidize organic contaminants to a more stable product in aqueous media at ambient temperature and pH.

Nitrophenols and their derivatives are widely used for preparing pharmaceuticals, synthetic dyes, and pesticides [19]. Nitrophenols are toxic and refractory organic compounds that can affect human and ecosystem health [18,20–22]. Due to the high stability and solubility of 4-nitrophenol (4-NP) and 2-nitrophenol (2-NP) in water, these compounds persist in the environment for a longer time (half-life) than other nitrophenol isomers (cf.

Figure 2) [23]. Thus, nitrophenols should be removed from water and wastewater, where adsorption-based removal is a relatively facile and effective strategy for meeting this goal. Porous materials are generally thought to be good adsorbents that play an important role in improving adsorptive separations or purification [24].

**Figure 2.** Molecular structures of 2-, 3-, and 4-nitrophenol isomers.

Various methods [25–43] are used for the detection of nitrophenols (Table S1; Supporting Materials). Among these methods, electrochemical strategies aimed at chemical transformation of the nitrophenols via oxidation or reduction processes are feasible. The utility of eliminating nitrophenols via an adsorption process, combined with reduction/oxidation of these harmful compounds, offers a potential remediation strategy. Thus, to remove nitrophenols from aqueous media, Ag NPs were deposited onto PANI-citric acid (Ag@P-CA) to investigate the electrochemical ability of such PANI-based electrocatalysts for the controlled oxidation of the adsorbed 2-NP and 4-NP isomers.

## 2. Results and Discussion

### 2.1. Thermogravimetric Analysis (TGA) and Atomic Absorption Spectroscopy (AAS)

TGA is a thermal analytical method that monitors the weight of a sample over time as the temperature varies [44]. Combined with AAS, TGA can be used to estimate the weight content (%) of various components in modified PANI-based composites. Upon measuring the amount of Ag content in each composite with AAS, the weight (%) of other components (water, PANI, dopant acid, and aniline oligomers) can be estimated based on the weight loss profiles of the modified PANI-based composites.

As shown in Figure S1 (cf. Supplementary Materials), PANI is characterized by three thermal events. The first relates to dehydration or loss of solvent in a thermal range of 25–150 °C. The second event is assigned to the loss of dopant and/or oligomers in the thermal range of 150–400 °C, and the third event is the complete decomposition of the polymer backbone above 400 °C. The three weight loss events are common features of doped PANI [45]. According to Figure S1 and Table 1, citric acid (CA) undergoes variable weight loss (multiple bands) that relate to the thermal decomposition of CA involving several intermediate products via dehydration and decarboxylation reactions [46]. In contrast, the formation of CA-modified PANI (P-CA) composites reveals a change in the thermal profile of CA when it is bound to PANI, where a single band is assigned to its decomposition. This alteration in the behavior of bound CA relative to free CA molecules provides support for a change in the chemical environment of CA as the dopant and modifier of PANI. After the deposition of Ag NPs onto P-CA (Ag@P-CA), the band attributed to CA in Ag@P-CA became less pronounced, which could be attributed to the charge neutralization of P-CA solution before the addition of the AgNO$_3$ solution. Incorporating CA into the structure of PANI (Ag@P-CA) resulted in greater thermal stability (as demonstrated by the attenuated weight loss over a wider temperature range) versus Ag@PANI.

**Table 1.** Estimated composition (wt.%) of different PANI-based and organic components of the composite based on TGA and Ag content based on AAS results. [1]

| Composite | Weight (%) of Various Components TGA Thermal Events | | | AAS Result | Aniline: CHI Feed Ratio | PANI: Non-PANI Ratio |
| | First Water | Second non-PANI | Third PANI | Ag Content | | |
|---|---|---|---|---|---|---|
| Ag@PANI | 1.9 | 24.2 | 67.7 | 6.2 | ∞ | 2.8 |
| Ag@P-CA | 0.51 | 8.3 | 86.1 | 5.1 | ∞ | 10.4 |

[1] Note: A description of the mass-balance for the composites can be found in the Supplementary Material in Section S1. Non-PANI content is referred to as the mixture of doping acid and aniline oligomers, if such species are present.

The combined use of AAS and TGA data in Table 1 for estimating the composition of Ag@P-CA and Ag@PANI reveals the presence of PANI oligomers in PANI-modified Ag NPs (Ag@PANI), where more than 20% non-PANI content is assigned to the oligomers in the structure of Ag@PANI. In comparison, Ag@P-CA includes 86.1% PANI and less than 10% non-PANI content, which is attributed to the remaining CA. This variation in the portion of non-PANI material was attributed to higher electrical conductivity for the Ag@P-CA than Ag@PANI, due to the effect of oligomers on reducing the electronic conductivity of PANI [47].

*2.2. FTIR Spectroscopy*

PANI and CA contain various functional groups and dipole moments (Figure 3) that can be structurally investigated with FTIR spectroscopy.

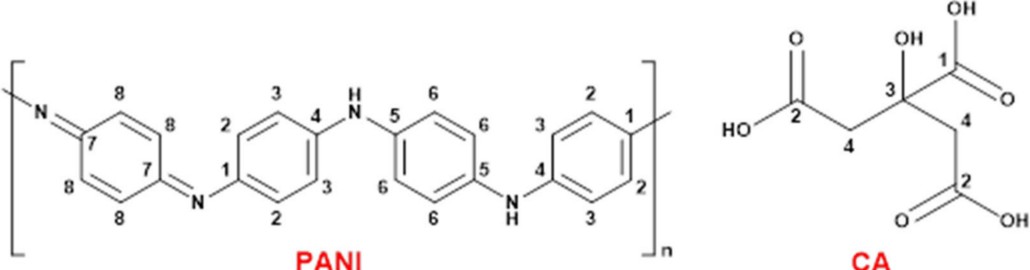

**Figure 3.** Molecular structures of PANI and CA and a carbon numbering scheme.

Figure 4 shows the region of 400–2000 cm$^{-1}$ of the FTIR spectra of modified PANI-based composites. PANI exhibits distinctive vibrational bands at 1586, 1509, 1308, 1242, 1166, and 833 cm$^{-1}$ [48]. The bands at 1586 and 1509 cm$^{-1}$ are assigned to the C=C and C=N stretching modes of vibration for the quinoid (-N=Q=N, where Q = quinoid ring) and benzenoid (-N-B-N-, where B = benzenoid ring) units, respectively. The band at 1308 cm$^{-1}$ is assigned to the C-N stretching mode of the benzenoid unit, whereas the band at 1166 cm$^{-1}$ originates from the bending mode of quinoid units of PANI (-N=Q=N-). The band at 833 cm$^{-1}$ is attributed to C-C and C-H stretching modes for the benzenoid units of PANI, which lend support for the occurrence of para-disubstituted ring systems [45].

The vibrational bands of PANI can be found in the PANI composites with and without Ag NPs and CA, which supports the successful synthesis of PANI by using CA as the doping agent and modifier. The CA signature at 1715 cm$^{-1}$ can be seen in the FTIR spectra of P-CA, due to the C=O vibrational band.

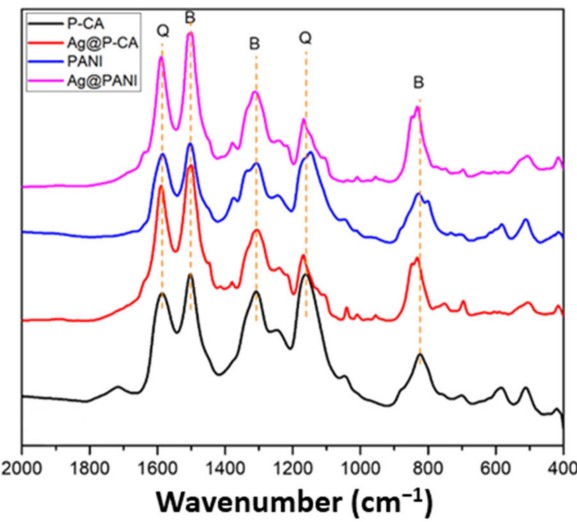

**Figure 4.** FTIR spectra of PANI-based composites. Orange lines represent characteristic peaks of PANI. Q and B represent the peaks assigned to quinoid (Q) and benzenoid (B) units, respectively.

### 2.3. X-ray Diffraction (XRD)

X-ray diffraction (XRD) is a non-destructive technique that provides information about the crystalline structure of materials with long-range order [49]. The diffraction patterns obtained for Ag NPs supported on the composite materials, along with XRD for the bare Ag NPs, are shown in Figure 5. The figure depicts the XRD pattern of Ag NPs deposited onto P-CA and PANI. Both composites displayed a similar pattern that includes diffraction peaks corresponding to planes of fcc crystal structure of metallic silver [50]. In addition, Figure 5 provides a comparison of the size of Ag NPs on PANI and P-CA composite supports. Ag deposited onto P-CA had an average size of 21.5 ± 5.2 nm, while Ag deposited onto PANI had an average size of 16.0 ± 3.2 nm. This trend provides evidence that changing the acid from HCl to CA did not change the size of Ag NPs deposited onto the composites.

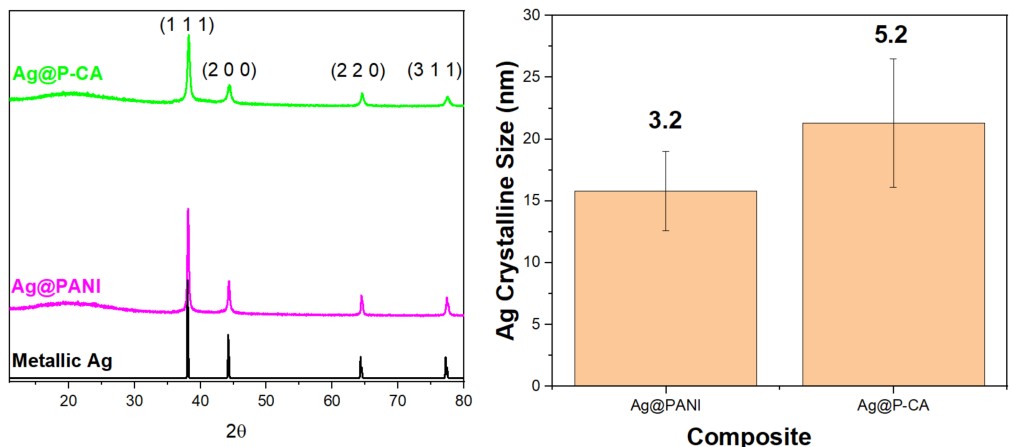

**Figure 5. Left panel**: XRD patterns of Ag@PANI-based composites and their characteristic diffraction profiles. Metallic Ag represents the calculated pattern of fcc Ag (RRUFF ID: R070463.9). **Right panel:** Comparison of the size and the homogeneity of the particle size of the Ag clusters in different composites based on the XRD results. The numbers above the columns indicate the standard deviation (nm). Four diffraction peaks were analyzed for each composite to obtain the Ag crystalline sizes and standard deviations. The microstrain effect and instrumental broadening were considered in the size calculations.

### 2.4. $^{13}$C Solid-State NMR Spectroscopy

The $^{13}$C solids spectra of PANI [51] were reported previously, where the results are summarized below. The PANI NMR spectrum in Figure 6 has seven spectral signatures at variable chemical shift ($\delta$; ppm) values, as follows: 115 (shoulder), 123, 128, 136, 140, 147, and 163 ppm. The $^{13}$C signatures at 123 ppm and 128 ppm and the shoulder at 115 ppm are assigned according to the scheme shown in Figure 3 for the various carbons C-2, C-3, and C-6, respectively. The lines at 136 ppm and 165 ppm arise from the C-8 and C-7 in the quinoid section of the PANI structure, respectively. The signature at 140 ppm was assigned to C-4 and C-5, and C-1 appears at 147 ppm [51].

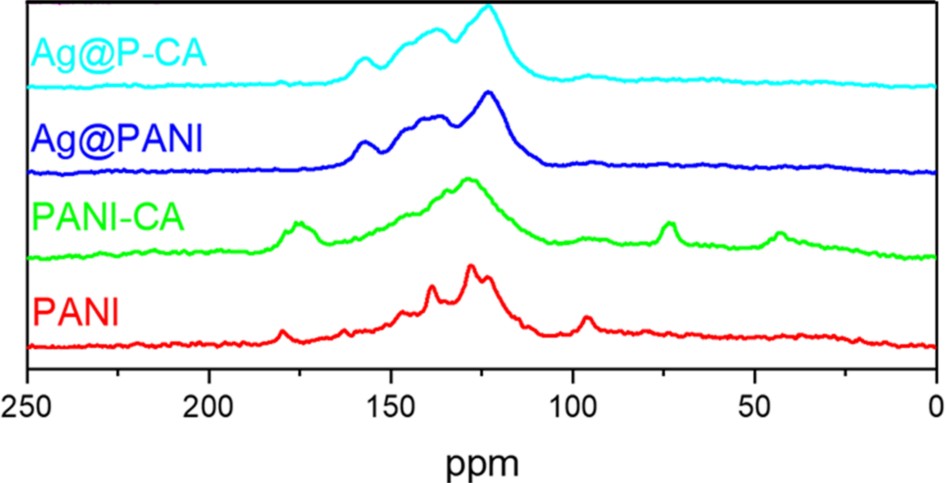

**Figure 6.** $^{13}$C CP/TOSS solid-state NMR spectra of PANI-based composites. The studies were performed at 7.5 kHz with a 5 µs $^1$H 900 pulse with a contact duration of 1.0 ms and a ramp pulse on the $^1$H channel.

In Figure 6, CA has four assigned characteristic spectral lines, as follows: 48 ppm (C-4), 78 ppm (C-3), 182 ppm (C-2), and 185 ppm (C-1) of CA, according to the numbering scheme in Figure 3 [52]. The CA signatures that appeared in the spectrum of P-CA vanished after the deposition of Ag NPs onto P-CA (Ag@P-CA). This disappearance is due to the pH adjustment before adding P-CA to the AgNO$_3$ solution and concur with the TGA results (Table 1). Interestingly, after deposition of Ag NPs onto P-CA and PANI, the PANI signature assigned to C-7 shifted upfield and became sharper. This shielding could be due to the interactions between the Ag NPs with the quinoid units, which affects the C-7 signature of PANI. The greater intensity of this band may relate to the oxidation of benzenoid (B) to quinoid (Q) units during the reduction process of Ag$^+$ to Ag NPs [53].

### 2.5. Dye Adsorption Study

The use of suitable dye adsorption probes can provide insight into the adsorption properties of the modified PANI-based composites. 2-NP was chosen as the target dye since this study aims to explore the electrocatalytical activity of Ag@P-CA for the oxidation of nitrophenols. The Langmuir (Equation S8) [54], Freundlich (Equation S9) [55], and Sips (Equation S10) [56] isotherm models have been used to characterize the adsorption isotherms (Figure S2) and the related adsorption parameters, according to the best-fit results for the isotherm models (cf. Section S3 in the Supplementary Materials). The Langmuir model assumes a homogeneous adsorbent surface with a monolayer adsorption profile, where adsorption at a given site is independent and of equal energy. The Freundlich model considers surface heterogeneities and attempts to incorporate the role of substrate–substrate interactions on the surface and the potential role of multilayer adsorption. The Sips model is a hybrid model that accounts for features of the Langmuir and Freundlich equations. The best-fit isotherm results for the adsorption profiles of 2-NP with different PANI-based composites at pH = 7 and 295 K are shown in Figure S2 (cf. Supplementary Materials).

The adj $R^2$ values indicate that the Sips model offers the best fit among the models (Langmuir, Freundlich, and Sips) evaluated, where the Sips isotherm parameters are listed in Table 2. The exponential term ($n_s$) in the Sips model accounts for surface heterogeneities, where values of $n_s$ that deviate from unity may indicate the dual role of adsorption sites attributed to PANI's B and Q sites of the polymer framework [45]. The deviation of $n_s$ from 1 ($n_s \neq 1$) for the composites indicate the heterogeneity of the adsorption sites, which can enhance the adsorption capacity ($Q_m$) for 2-NP for such composites.

**Table 2.** Best-fit isotherm parameters for the Sips model in a PBS-7 buffer solution at 295 K.

| Composites | Sips Isotherm | | | |
| --- | --- | --- | --- | --- |
| | $K_S$ | $Q_m$ ($\mu mol \cdot g^{-1}$) | $n_s$ | Adj $R^2$ |
| PANI | $0.00272 \pm 0.00029$ | $515 \pm 23$ | $1.56 \pm 0.24$ | 0.977 |
| Ag@P-CA | $0.00057 \pm 0.00036$ | $1520 \pm 371$ | $0.85 \pm 0.15$ | 0.981 |
| Ag@PANI | $0.00186 \pm 0.00035$ | $733 \pm 56$ | $1.25 \pm 0.21$ | 0.975 |

After the deposition of Ag NPs onto PANI, Ag@PANI was able to adsorb a greater amount of 2-NP versus PANI. This tendency highlights the role of Ag NPs as the active adsorption site for 2-NP. Ag@P-CA had twice the value of $Q_m$ versus Ag@PANI, which shows the successful modification of PANI with CA and its effect on the adsorption of 2-NP. The higher value of the monolayer adsorption capacity ($Q_m$) ensures the presence of the target molecule on the surface of the catalyst. While the value of $Q_m$ for Ag@P-CA is high, its binding affinity ($K_S$) is more than three times lower than that for Ag@PANI. The lower $K_S$ value ensures the detachment of the target species from the catalyst upon completion of the electrochemical reaction.

*2.6. UV-Vis Spectroscopy*

To evaluate the electronic properties of the synthetic composites, UV-Vis spectroscopy and XPS can be utilized to investigate the valence band spectra of modified PANI-based composites [57]. The UV-Vis spectra of PANI and P-CA are shown in Figure S3 (cf. Supplementary Materials). The spectra of the samples were scaled from 0 to 1 to normalize the effect of concentration on the optical absorption intensity.

The UV-Vis spectra of PANI have many vibrational bands, where support is provided herein for the characterization of PANI by assigning vibrational bands to the benzenoid and quinoid units of the polymer. Then, the presence of polarons and bipolarons, which are responsible for the electron conductivity of PANI [47], can be further investigated.

Depending on the preparation and/or processing of PANI, a typical absorption spectrum of emeraldine base contains two different absorption bands positioned between 315 and 360 nm and 610 and 650 nm [58–61]. After the acid doping of PANI, the protonated form of PANI (ES) in Figure 1 reveals new absorption bands at 400–430 nm [61]. The absorption in the 315–360 nm (Figure S3) region was assigned to a fundamental structural unit of the PANI chain and is frequently attributed to electron transitions from the benzenoid segments [59,62]. The absorption band at 400–430 nm [63] is frequently attributed to polaron transitions [64,65]. The neutral form of the quinoid diimine structure, [59] in conjugation with neighboring benzenoid units (cf. Figure 1), and its conversion to the protonated form are related to variations in absorption over the red spectral region (610–650 nm), attributed to polaron structure. The band at 610–650 nm tends to diminish as the EB sample is electrochemically reduced to the leucoemeraldine base form of PANI. This absorption band is replaced for the oxidized form (PG) in Figure 1 by an additional absorption band at a wavelength below 600 nm. In addition, an extra band was evident below 300 nm for PG, whereas EB lacks this vibrational band (cf. Figure 1).

PANI-CA appeared in the ES form due to the lack of characteristic bands of PG and the simultaneous appearance of an absorption band in the 400–430 nm region. Table 3 summarizes the UV-Vis spectra for PANI and P-CA, based on the results presented in

Figure S3 (Supplementary Materials). The wavelength ranges were also converted to the energy ranges in order to compare with the corresponding XPS results.

**Table 3.** Summary of the solid-state UV-Vis absorbance spectra of the PANI and P-CA based on the results shown in the Supplementary Materials (Figure S3).

| Highlighted Region | Spectral Range (nm) | Energy Range (eV) | Spectral Band Assignment | Samples without Relative Absorption Bands |
|---|---|---|---|---|
| Red (I, IV) [64,65] | 500–600 | 2.1–2.5 | PG form of PANI | P-CA |
| Blue (V) [58–61] | 610–650 | 1.9–2.0 | Quinoid units | none |
| Yellow (II) [59,62] | 315–360 | 3.4–3.9 | Benzenoid units | none |
| Green (III) [63–65] | 400–430 | 2.9–3.1 | Polarons | none |

Note: Samples with relative absorption bands are those that showed the relative absorption band in their UV-vis spectra.

## 2.7. X-ray Photoelectron Spectroscopy (XPS)

Neutral conjugated polymers are usually low-conductivity semiconductors. In Figure S4 (Supplementary Materials), the valence band XPS profiles of PANI and P-CA are shown, which are calibrated to the C 1s (adventitious carbon) spectrum at 284.8 eV to enable study of their valence band structure attributes [66]. Based on Figure S4 (cf. Supplementary Materials), the band gap energy of each composite can be estimated. Thus, the lower binding energy edge represents the lower binding energy required to stimulate an electron to move from the valence band to the conduction band. In the case of insulating and semiconductor polymers, the lowest binding energy edge of the spectrum is not $E_F$. In nonconductors and semiconductors, the Fermi level is ill-defined and is often believed to be in the middle of the band gap [67]. A summary of the XPS results and the related parameters is provided in Table 4.

**Table 4.** Summary of XPS results based on the spectra in the Supplementary Materials (Figure S4).

| Sample | Estimate of Half the Bandgap Energy (eV) | Estimate of Bandgap Energy (eV) |
|---|---|---|
| PANI | 2.3 | 4.6 |
| P-CA | 0.2 | 0.4 |

P-CA has the lowest binding energy (0.4 eV), which can be accounted for due to the oxidation state of P-CA (ES; PANI with its highest conductivity). The presence of CA (doping agent) in its pores (based on AAS) results in conductivity features that are characteristic of a semi-metallic composite. For comparison, the lowest binding energy edge of gold is 0.055 eV [68].

## 2.8. Cyclic Voltammetry (CV)

CV is a robust and widely used electrochemical method for studying the reduction and oxidation processes of chemical species. CV aides in the study of electron transfer-initiated chemical processes such as catalysis [69].

The CV profiles of PANI and its modified composites depend on the nature of the counter ion of the acid dopant, the degree of protonation, and the oxidation state of PANI [70–72]. Based on Figure 7, three oxidation states of PANI (cf. Figure 1) can be observed: LE, EB, and PG. LE was oxidized to EB at 200 mV, and EB was oxidized to PG at 650 mV. The reduction reaction from PG to LE (second sweep) with a low $\Delta E_p$ showed the facile electron transfer between PANI chains [71]. After Ag NPs were deposited onto the composites, a new signature near 200 mV appeared, which was attributed to the oxidation of Ag (cf. Figure 7) [73].

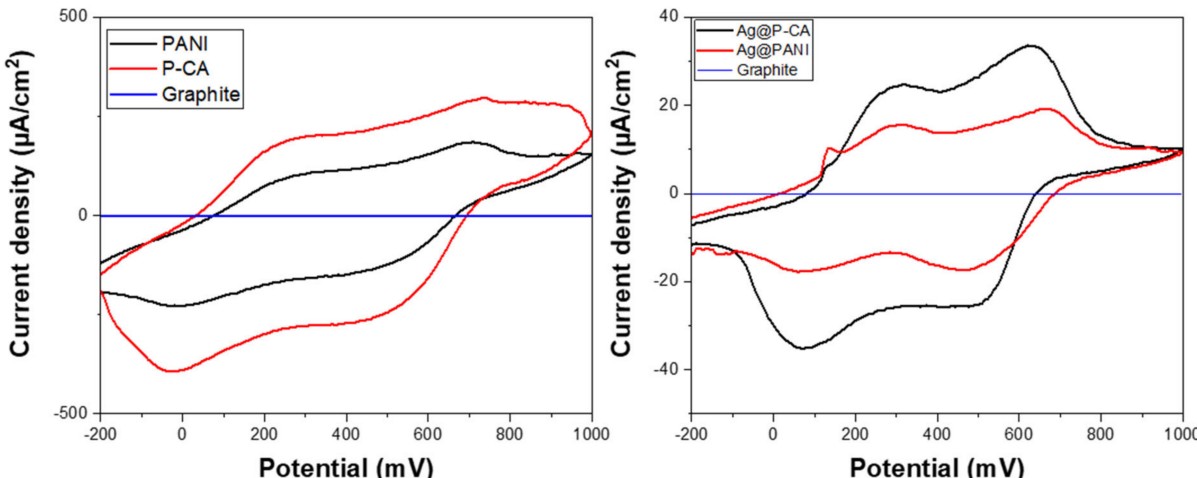

**Figure 7.** Cyclic voltammetry characterization of PANI-based composites in acidic solution (pH = 1). Left panel shows PANI materials without Ag NPs and the right panel shows PANI materials with Ag NPs. The CV analysis of synthesized ternary composites that contain Ag NPs was carried out at a scan rate of 20 mV·s$^{-1}$ and a potential range from −200 to 1000 mV using the three-electrode system. The results in both panels are referenced to a pure graphite paste electrode material.

To compare the electroactivity of the composites, CVs were measured in PBS-7 solution at a scan rate of 20 mV·s$^{-1}$ using potassium ferricyanide as a redox marker (Figure 8). The absence of a significant redox signal for a plain carbon paste electrode material without a catalyst (Ag NPs) was noted. CVs of Ag@PANI and Ag@P-CA had two peaks, which are assigned to oxidation and reduction in iron species (Fe (II)/Fe (III)). Based on Figure 8B, Ag@P-CA revealed a lower $\Delta E_P$ (highest electron transfer rate) and higher $i_p$ (higher distribution of the target analyte on the surface of the electrode material), as compared with Ag@PANI. These results provide evidence for the successful modification of Ag@PANI, using CA as the doping acid and modifier.

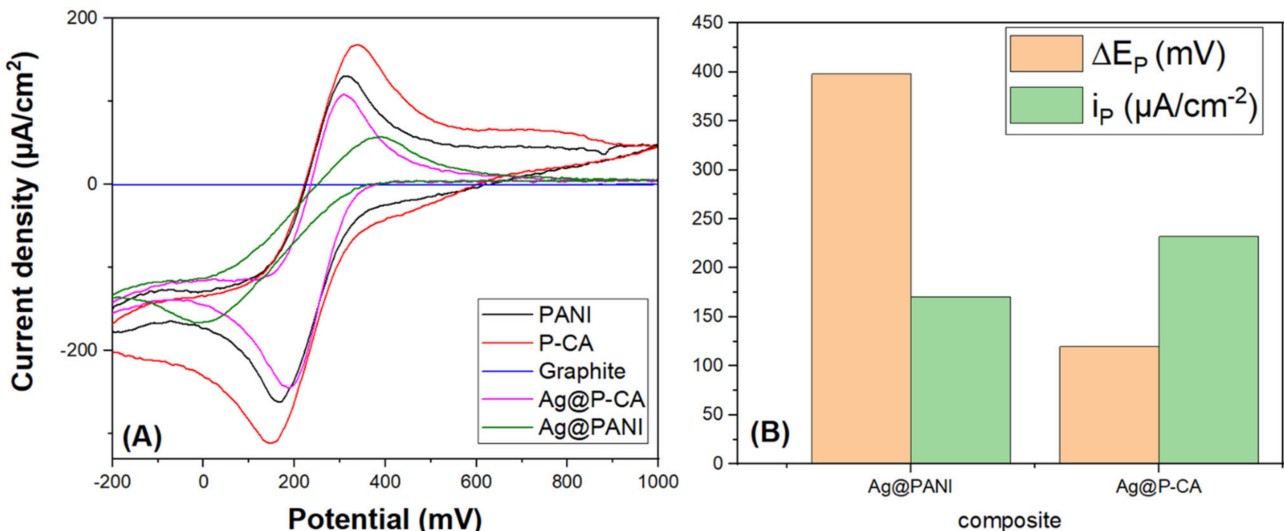

**Figure 8.** (**A**) CV analysis of modified PANI-based composites in a PBS-7 buffer solution containing 0.1 M of $K_3Fe(CN)_6$. (**B**) $\Delta E_P$ and maximum $i_p$ derived from (**A**). The CV analysis of synthesized ternary composites was carried out at a scan rate of 20 mV·s$^{-1}$ and a potential range from −200 to 1000 mV using the three-electrode system.

By employing Ag@PANI and Ag@P-CA as working electrode materials, CV experiments were performed in the presence of 2-NP and 4-NP (Figure 9). The $i_p$ assigned to the oxidation of Ag NPs and 2-NP can be seen in all the graphs, regardless of the type

of the working electrode material. As a result, both the working electrode materials are sensitive to the presence of 2-NP and 4-NP in the solution. The nonlinear change appears in the peak current density of Ag@PANI-modified electrode material by altering 2-NP and 4-NP concentrations (Figure 9A,C), which suggests the insensitivity of Ag@PANI to the concentration of 2-NP and 4-NP in the solution. In contrast to Ag@PANI, peak current density of Ag@P-CA increased linearly with increasing dye concentrations from 200 µM to 1000 µM, which resulted in its sensitivity to the concentration of 2-NP and 4-NP in the solution. (cf. Figure S6, Supplementary Materials). Recent studies [27–30,32,37,40–42,74] have demonstrated that the LOD for 2-NP and 4-NP resides in the nM range (Table S2).

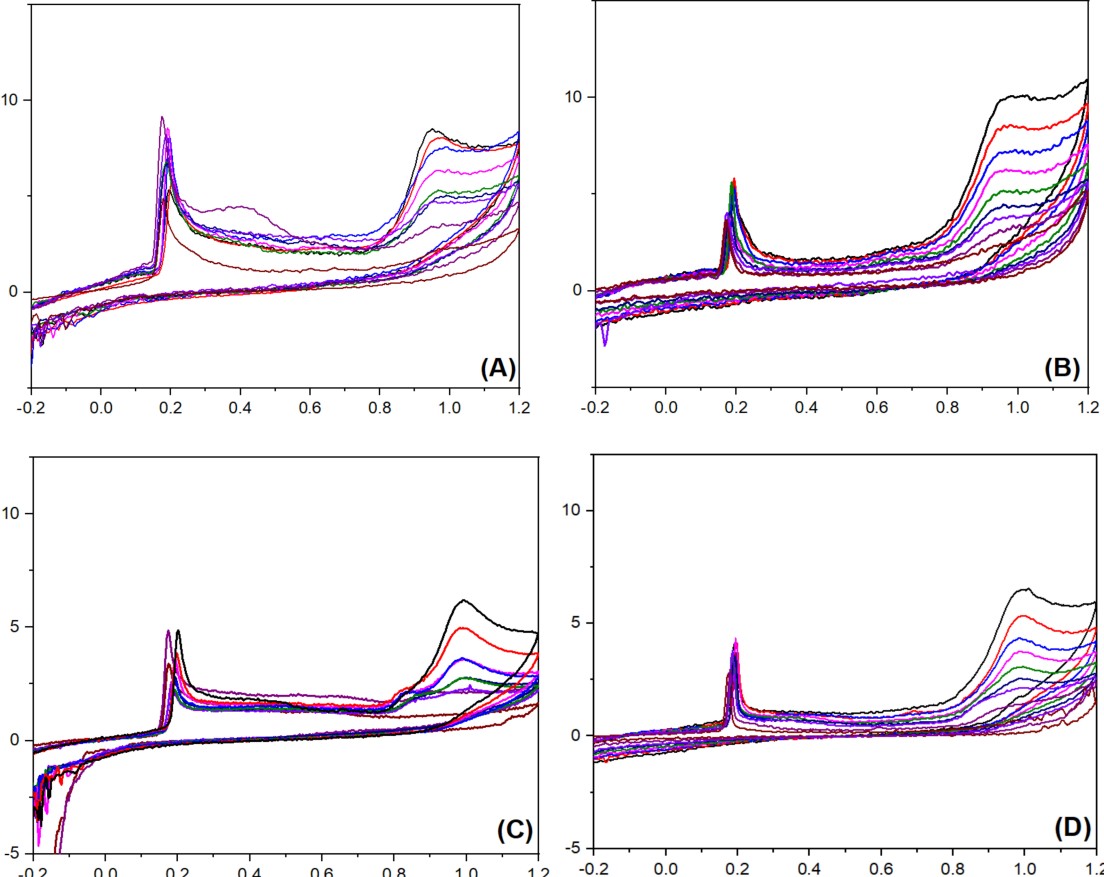

**Figure 9.** Detection of 2-NP with (**A**) Ag@PANI and (**B**) Ag@P-CA electrode materials. Detection of 4-NP with (**C**) Ag@PANI and (**D**) Ag@P-CA electrode materials. The Y-axis represents current density ($\mu A.cm^{-2}$), and X-axis represents the potential (V). The CV analysis of the composites was carried out at a scan rate of 50 mV·s$^{-1}$ and a potential range from −600 to 1200 mV using the three-electrode system in PBS-7 buffer solution (pH = 7) and 298 K.

The concentration sensitivity of the modified working electrode materials suggests that Ag@P-CA could be effective as a sensor material as well as an electrocatalyst for the detection/oxidation of 2-NP and 4-NP. The CV profiles indicate two peak currents. The first peak at 0.2 V is assigned to the oxidation of Ag NPs, and the second (between 0.91 and 1.02 V) was assigned to the oxidation of 2-NP and 4-NP. Both peak currents increased linearly with incremental 2-NP (or 4-NP) concentration. This trend demonstrates a response of the Ag NPs as the catalyst for the oxidation of 2-NP (or 4-NP). The role of Ag NPs was also observed by examining 2-NP adsorption isotherm parameters in Table 2. After the deposition of Ag NPs onto PANI, $Q_m$ increased while $K_S$ decreased dramatically, which is favorable for electrocatalyst applications. The shape of the voltammogram is related to the surface coverage of the electroactive species on the working electrode material. Greater surface coverage concurs with a higher oxidation peak [73]. The CV experiments resulted in

no selectivity between nitrophenol isomers using Ag@P-CA as the electrocatalyst, while the RSD remained lower than 3.2% for all of the $C_{2\text{-NP}}$-to-$C_{4\text{-NP}}$ ratios illustrated in Figure 10.

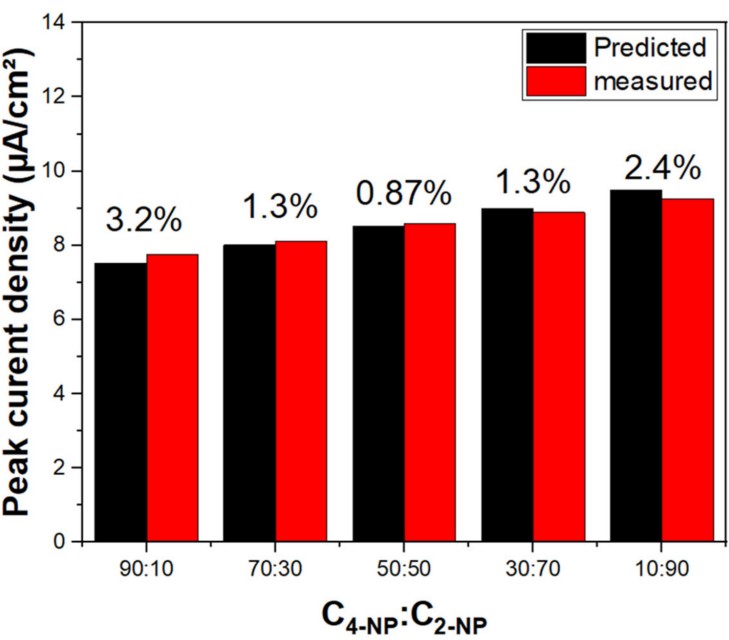

**Figure 10.** Nonselective detection of 2-NP and 4-NP with Ag@P-CA at different concentration ratios ($C_{4\text{-NP}}$: $C_{2\text{-NP}}$). The numbers above columns represent the error (%) based on the measured current densities compared to the predicted ones. Predicted current densities are based on the sum of calibration curves of 2-NP and 4-NP. The experiments were performed in PBS-7 buffer solution.

In general, the rate of various processes, such as (a) mass transfer, (b) electron transfer, (c) chemical reactions before or following the electron transfer, and (d) other surface interactions such as adsorption, desorption, or crystallization, influence the current (or electrode reaction rate), as illustrated in Figure 11 [69]. Thus, in order to increase the current density, the electron transfer needs to be increased by optimizing the electrochemical properties of the composites. Furthermore, a modified electrode material requires high surface coverage of the targeted analyte with low binding affinity relative to its precursors [69]. Therefore, to increase the electrode reaction rate, all these steps require optimization. Mass transfer and chemical reactions before or following the electron transfer are independent of the properties of the electrode material. However, modification of the electrode material can lead to optimization of the electron transfer, adsorption, and desorption steps of the electrocatalytic process.

In general, a higher rate of electron transfer results in a higher electrode reaction rate. Higher electrical conductivity can be obtained in conjugated polymers (such as PANI) under the conditions listed here: (i) low π-π stacking distances (higher interchain charge transfer is integral to avoid charge localization), (ii) planar backbone structure, and (iii) highly homogeneous crystalline orientation (no orientation anisotropy and better percolation pathway) and the presence of bridging polymers [47]. Two outcomes from the results provide support that the electronic conductivity of PANI increased after modification with CA: (i) XPS results showed that P-CA had a bandgap that was more than 11 times lower than the bandgap of PANI (Table 4), and (ii) $\Delta E_p$ was lower for Ag@P-CA, compared to Ag@PANI (Figure 8). There is experimental evidence that supports why Ag@P-CA should have a higher conductivity, as compared with Ag@PANI. The ratio of PANI: non-PANI for Ag@P-CA was 10.42, whereas for Ag@PANI, this ratio equaled 2.79 (Table 1). The crystalline domains of the polymer are formed by a high degree of π-orbital overlap, whereas amorphous regions are formed by weak interactions of less-ordered polymer chains such as oligomers. By considering the non-PANI content as mainly oligomers, the crystallinity level of Ag@P-CA can be considered higher than that for Ag@PANI [75]. The

impact of CA on the crystallinity of PANI can be attributed to micelle formation in the media during the synthesis. Since the ammonium peroxydisulphate (APS) employed as the oxidant is hydrophilic, the polymerization occurs mainly at the water–micelle contact region with PANI chains forming inside the micelle (cf. Figure S5, Supplementary Materials). The sharp vibrational band at 833 cm$^{-1}$ in the FTIR spectra of P-CA and Ag@P-CA indicates the para-disubstitution of the rings. The polymer chains containing para-linked aniline units joined in the head-to-tail mode can delocalize the charge along with the conjugated system, lowering its energy and extending electron transfer along the polymer chains [76].

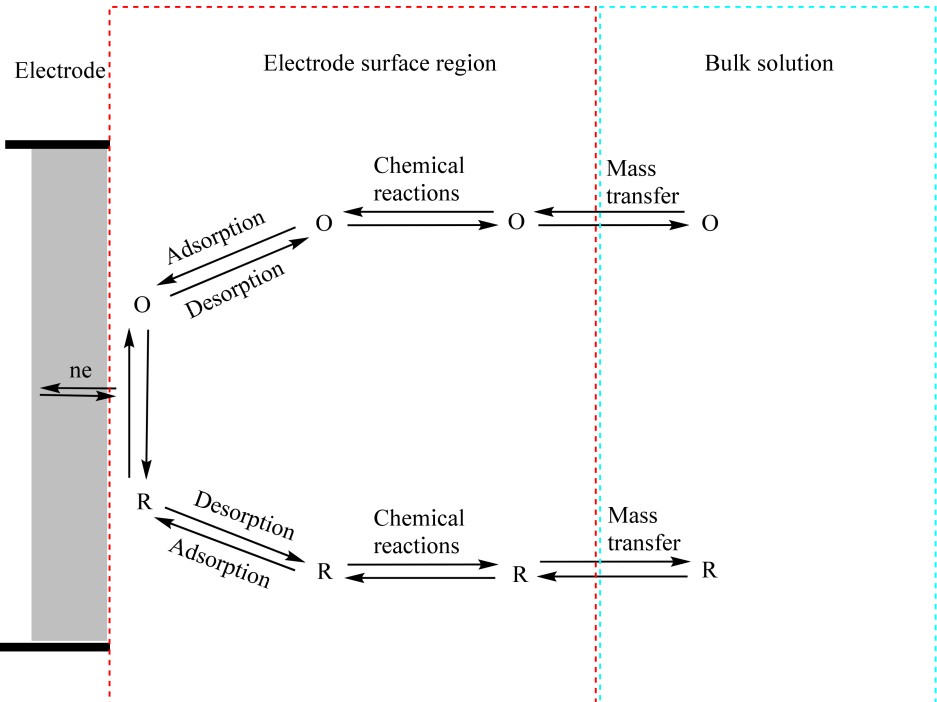

**Figure 11.** Illustrative pathway of a generalized electrode reaction. O and R represent the oxidized and reduced forms of the chemical species, respectively. Adapted with permission from Ref. [69].

An electrocatalyst requires high capacity for the adsorption of the reactant and its facile desorption from the surface of the electrocatalyst. PANI modified with Ag NPs and CA had the highest adsorption capacity ($Q_m$) for 2-NP and the lowest binding affinity ($K_S$). The role of Ag NPs as the adsorbent for dyes was previously reported [77]. On the other hand, greater adsorption of 2-NP by Ag@P-CA over Ag@PANI may relate to the effect of CA on the morphology of the PANI composite discussed above (cf. Section 1).

The high electronic conductivity of Ag@P-CA along with its optimized adsorption and desorption properties, as compared to Ag@PANI, indicate that Ag@P-CA is a candidate for electrocatalyst applications [69]. Based on the CV results in the presence of nitrophenols, a linear relationship was established between the concentration of nitrophenols and current density, which suggests that Ag@P-CA works in a dual fashion, both as an electrocatalyst and as a sensor material for nitrophenols.

## 3. Materials and Methods

### 3.1. Materials

All chemical reagents were analytical grade. Aniline, a brown liquid with a density of 1.022 g·cm$^{-3}$, was purchased from Sigma-Aldrich (Oakville, ON, Canada) with an ACS reagent purity grade of 99.5%. Silver nitrate (AgNO$_3$), sodium borohydride (NaBH$_4$) ($\geq$ 95%), and potassium bromide (KBr) ($\geq$ 99.5%) were purchased from BDH Chemicals (Toronto, ON, Canada). Ammonium peroxydisulphate (98% purity grade) and citric acid monohydrate ($\geq$ 99.0%) were purchased from Sigma-Aldrich (Oakville, ON, Canada).

Hydrochloric acid (12.1 N) and sodium hydroxide (NaOH) were purchased from Fischer Scientific International Inc. (Ottawa, ON, Canada). 2-nitrophenol was purchased from Eastman Kodak Company (Burnaby, BC, Canada) and 4-nitrophenol (4-NP) (99%) was purchased from Sigma-Aldrich (Oakville, ON, Canada) and recrystallized twice from purified water; then, it was dried in the oven for several days at 50 °C. All solutions were prepared using high purity Milli-Q water with a resistivity of 17–18 MΩ·cm$^{-1}$.

### 3.2. Methods

#### 3.2.1. Synthesis of Ag@Polyaniline-Citric Acid Composite

Composites were prepared using a typical in situ reduction method [78]. To prepare Ag@P-CA, 1.47 g of APS was dissolved in 20.0 mL of 0.100 M CA. Then, 0.150 mL of aniline was added to 25.0 mL of 0.100 M CA and agitated for 0.5 h. Following that, 20 mL freshly prepared aqueous APS solution was added dropwise into the reaction mixture at a rate of 1 drop·s$^{-1}$. The pH of the solution was adjusted to 6.50 using NaOH and HCl after 24 h of reaction. To eliminate excess APS, the polymer was filtered using a Whatman filter (125 mm diameter, 8 μm pore size) by several washings with 100 mL of millipore water to remove the excess APS, followed by 50 mL of acetone to remove the oligomers. The polymer composites were then partially dried in an oven at 50 °C for 3 h, following by grinding and sieving. Before dispersing the polymer composite (P-CA) in 125 mL of 5.00 mM AgNO$_3$ solution, some of the P-CA material was collected for further physicochemical characterization. The solution was maintained in an ice bath with constant stirring. Then, 375 mL of 10.0 mM NaBH$_4$ was added dropwise for 2 h to the solution, along with stirring for 1 h. Finally, the product was rinsed with 100 mL of millipore water and collected using centrifugation and an oven-drying procedure for 24 h at 50 °C. Then, the resultant material was crushed into a powder form, sieved (125 μm mesh size), and stored for future use.

#### 3.2.2. Fabrication of Electrodes

A total of 10.0 mg composite was mixed with 100 mg of fine-quality graphite flakes and co-ground for 0.5 h. Then, 10.0 μL of mineral oil was added to the mixture to form a homogeneous paste. A total of 2 mg of paste was separated from the whole and filled at the bottom of the glass capillary (2 mm diameter), where a clean copper wire was used for the connection.

#### 3.2.3. Preparation of Phosphate-Buffered Saline (PBS-7) Solution

The solution was produced for 1.00 L of PBS-7 with 800 mL millipore water combining 2.00 g NaCl, 200 mg KCl, 1.44 g Na$_2$HPO$_4$, and 245 mg KH$_2$PO$_4$. The pH was adjusted to 7 with the appropriate amount of HCl and/or NaOH; then, the volume was increased to 1 L with millipore water to yield a solution with a phosphate concentration of 11.94 mM.

#### 3.2.4. Preparation of Nitrophenol Solutions for Cyclic Voltammetry Study

Nitrophenol solutions with concentrations that range from 10.00 to 1000 μM were made using PBS-7 (11.94 mM phosphate) solution as media.

#### 3.2.5. Thermogravimetric Analysis (TGA)

Thermogram profiles of the compounds were obtained using a TGA instrument (Q500 TA Instruments). Samples were heated in open aluminum pans at 25 °C and allowed to equilibrate for 5 min before heating at a scan rate of 5 °C·min$^{-1}$ up to 500 °C with a balanced purge flow of 10.0 mL·min$^{-1}$.

#### 3.2.6. Atomic Absorption Analysis (AAS)

The content (%) of Ag NPs in each composite was measured using SPECTRAA 55 FAAS instrument. The composites were dissolved in nitric acid and diluted to achieve the Ag(I) optimum working range of 0.02–10 μg·mL$^{-1}$ for the wavelength of 328.1 nm and

a slit width of 0.5 nm. The lamp current was 4 mA, and the fuel was acetylene with air serving as the oxidizer source.

### 3.2.7. Fourier Transform Infrared (FTIR) Spectroscopy

FTIR spectra of powdered samples were obtained with a Bio-Rad FTS-40 spectrophotometer in reflectance mode. Solid samples were prepared by co-grinding polymers (5 mg) with pure spectroscopic grade KBr (50 mg). The DRIFT (Diffuse Reflectance Infrared Fourier Transform) spectra were obtained at 295 K with a resolution of 4 cm$^{-1}$ over the 400–4000 cm$^{-1}$ region.

### 3.2.8. X-ray Diffraction (XRD)

A Rigaku Ultima IV X-Ray Diffractometer with a Cu source (K$_{\alpha 1}$: 1.54060 Å and K$_{\alpha 2}$: 1.54443 Å), Cross-Beam Optics (CBO), and a Scintillation Counter detector were used for the powder XRD studies. The measurements were taken with the multipurpose attachment using the para focus method. The K$_{\beta}$ diffraction components were removed by using a Ni foil filter. The 2θ was checked with the Si powder sample (220) plane at 47.28° and scanned from 10° to 80° with a step size of 0.02° and a scan rate of 2° per minute for all samples with an exposure time of 0.6 s·step$^{-1}$.

### 3.2.9. $^{13}$C Solid-State NMR Spectroscopy

A Bruker Avance III HD spectrometer running at 125.77 MHz ($^{1}$H frequency at 500.13 MHz) and a 4 mm DOTY CP-MAS probe were used to conduct $^{13}$C solid-state NMR experiments. The $^{13}$C CP/TOSS (Cross Polarization with Total Suppression of Spinning Sidebands) studies were performed at 7.5 kHz with a 5 μs $^{1}$H 900 pulse, a contact duration of 1.0 ms, and a ramp pulse on the $^{1}$H channel. Using a 2 s recycle delay, 1700–4400 scans were collected for various samples. All spectra were recorded on a $^{13}$C channel utilizing 50 kHz SPINAL-64 with $^{1}$H decoupling during acquisition. The chemical shifts of $^{13}$C were referenced to adamantane at 38.48 ppm as the external reference (low field signal).

### 3.2.10. Solid-State UV-Vis Spectroscopy

The solid-state UV-Vis spectra of the samples were measured using Pistar-180 UV-vis/Circular Dichroism spectrometer from Applied Photophysics (Leatherhead, UK) in absorbance mode. The monochromator range (200–800 nm) was chosen for this study. The composites were dissolved in N-methyl-2-pyrrolidone (ca. 700 ppm) and filtered using Whatman filter papers (125 mm diameter, 8 μm pore size). A thin layer of sample solution was spin-coated onto quartz plates at variable spinning speed (<40 rpm) and time.

### 3.2.11. X-ray Photoelectron Spectroscopy

All X-ray photoelectron spectroscopy (XPS) measurements were collected using a Kratos (Manchester, UK) AXIS Supra system at the Saskatchewan Structural Sciences Centre (SSSC). This system is equipped with a 500 mm Rowland circle monochromated Al K$_{\alpha}$ (1486.6 eV) source and a combined hemispherical analyzer (HSA) and spherical mirror analyzer (SMA). A spot size of a hybrid slot (300 × 700 μm) was used. The valence band XPS spectra were collected in the 5–30 eV binding energy range in 0.05 eV steps with a pass energy of 40 eV. The XPS was calibrated to the C 1s spectrum at 284.8 eV. A thin sample layer was prepared by spin coating the gold-coated silicon plates (prewashed with ultrasonication in a dichloromethane bath) at variable spinning speed (<40 rpm) and time.

### 3.2.12. Cyclic Voltammetry Analysis

Electrochemical studies were carried out using a three-electrode configuration using a PGZ 301 Dynamic-EIS Voltammetry (VoltaLab) voltammeter (London, ON, Canada) with a model number of R21V008; modified carbon paste, Ag/AgCl (saturated with KCl), and Pt wire were used as working, reference, and counter electrodes, respectively. For investigating the response of the electrode materials to the concentration of nitrophenols,

one electrode was used for all the cycles at different concentrations, where the first scan was omitted from the presented results.

### 3.2.13. Dye Adsorption Study

The optical absorption spectrum was recorded by a Varian Cary 100 scan spectrometer equipped with a solid sample accessory in transmission mode between 300 and 800 nm. The dye adsorbate used for this study was 2-NP ($\lambda_{max}$ =416 nm, $\varepsilon$ = 2.13 mM$^{-1}$·cm$^{-1}$ in PBS-7 buffer solution). A calibration curve was built to calculate the molar absorptivity of 2-NP at its $\lambda_{max}$ to establish the operating range of concentrations ($R^2$ = 0.998). A total of 5.00 mM 2-NP stock solution was made by dissolving 0.348 g of 2-NP in 0.500 L of PBS-7 buffer solution.

## 4. Conclusions

Herein, composite materials were prepared by immobilizing Ag NPs onto polyaniline (PANI) as a semi-conductive support for electrochemical oxidization of 2-nitrophenol and 4-nitrophenol. PANI was modified with citric acid to enable optimization of the nitrophenol adsorption and desorption. The structure and electron transfer properties of modified PANI were characterized via XRD and CV studies. The composite, Ag-P-CA, is more conductive than PANI, which leads to facilitation of electrochemical transformations of nitrophenol, where it was shown that Ag@P-CA can be used as an electrocatalyst and as a sensor material. However, the LOD of Ag@P-CA for detection of 2-NP and 4-NP has some limitations. The latest studies achieved the LOD for the detection of 2-NP and 4-NP in the nM range (Table S2), while the lowest LOD that was achieved in this study was 155 μM (Figure S6). Therefore, optimizing the LOD becomes necessary to reach nM detection levels in the future studies.

**Supplementary Materials:** The following supporting information can be downloaded at: https://www.mdpi.com/article/10.3390/catal13030465/s1, Figure S1: First derivative TGA of modified PANI-based composites and their precursors, Figure S2: 2-NP adsorption study of modified PANI-based composites and their precursors and fitted isotherms at 295 K and pH = 7 in PBS buffer solution, Figure S3: Solid-state UV-Vis spectra of modified PANI-based composites and its precursors, Figure S4: Valence band XPS of modified PANI-based composites, Figure S5: Illustrated view of the role of organic acids in the polymerization of aniline, and a supplementary discussion of XRD, TGA, and dye adsorption results, Figure S6: Calibration curves for the detection of 2-NP and 4-NP with Ag@P-CA modified electrode material, Table S1: Recent examples of literature studies for the detection of nitrophenols, and Table S2: Examples of recent studies on the electrochemical detection of nitrophenols.

**Author Contributions:** Conceptualization, L.D.W. and R.S.; methodology, L.D.W. and R.S.; validation, L.D.W. and R.S.; formal analysis, M.K.; investigation, M.K.; resources, L.D.W. and R.S.; data curation, M.K.; writing—original draft preparation, M.K.; writing—review and editing, L.D.W., R.S. and M.K.; visualization, M.K.; supervision, L.D.W. and R.S.; project administration, L.D.W.; funding acquisition, L.D.W. All authors have read and agreed to the published version of the manuscript.

**Funding:** This research was funded by the Government of Canada through the Natural Sciences and Engineering Research Council of Canada as a Discovery Grant (RGPIN 04315-2021) awarded to LDW. MK acknowledges the partial support provided by the University of Saskatchewan through the award of a Graduate Teaching Fellowship (GTF).

**Data Availability Statement:** Not applicable.

**Acknowledgments:** The Saskatchewan Structural Science Centre (SSSC) is acknowledged for providing facilities to conduct this research. Funding from Canada Foundation for Innovation, Natural Sciences and Engineering Research Council of Canada and the University of Saskatchewan support research at the SSSC.

**Conflicts of Interest:** The authors declare no conflict of interest.

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
