# Peer review of "Electrocatalytic Oxidation of Nitrophenols via Ag Nanoparticles Supported on Citric-Acid-Modified Polyaniline"

_catalysts, doi:10.3390/catal13030465_

Round 1

Reviewer 1 Report

In this article the authors have synthesized by an in-situ reduction method, polyaniline modified with citric acid (P-CA) and Ag nanoparticles modified with P-CA (Ag@P-CA). The physical and chemical properties of P-CA and Ag@P-CA were compared with unmodified polyaniline (PANI) and PANI-modified Ag nanoparticles (Ag@PANI). The results of the characterization by X-ray photoelectron spectroscopy showed for the potential utility for the optimization of the adsorption/desorption and electron transfer steps during the electrochemical oxidation of nitrophenols. Ag@P-CA was used as an electrocatalyst by modifying polyaniline with Ag nanoparticles and citric acid, for the oxidation of 2-nitrophenol and 4-nitrophenol. These nanomaterials can significantly contribute to the identification of a strategy for detection and electrocatalytic remediation of nitrophenols and water.

The method of synthesizing these nanomaterials is original and can be used in various fields and industries. The authors have definitely proven a good knowledge and an important experience in the use of surface characterization methods.

I believe that the article could be improved if the authors take into account several aspects:

1. The introduction is clear, short and to the point. Describe very well the properties of the nanomaterials of interest. However, it seems essential  to provide some examples from the specialized literature of synthesizing such modified nanoparticles used as electrocatalysts or other ways of detecting nitrophenols. 1-2  review paragraphs on this topic would improve the introduction.

2. Figure 4 has two images A and B, but I don't see any difference and they are not mentioned in the text distinctly. Please verify or clarify.

3. In section 2.5., 1-2 citations would be useful to reinforce the interpretation of the parameters.

4. Why did you use CV as an electrochemical characterization method and not EIS, it is very suitable for evaluating the resistance to electron transfer.

5. There are certain inappropriate expressions that appear in several places. For example "current density" line 341, 375 etc.

6. In section 2.8. CV would be useful to show the variation of the intensity of the oxidation current with the increase in the concentration of nitrophenols, possibly a calculation of the detection limit (if possible).

7. In the conclusion section add some aspects regarding future research perspectives and possibilities of using these synthesized nanoparticles in several fields.

8. There are some phrases in the text that are too complicated and difficult to read. If possible, simplify the sentences to make it easier to read.

9. Approximately half of the references are quite old. I recommend replacing some with newer references, at least from the last 10 years.

Author Response

Authors’ Response to Reviewer Comments on MS ID:  catalysts-2210212

Reviewer #1

In this article the authors have synthesized by an in-situ reduction method, polyaniline modified with citric acid (P-CA) and Ag nanoparticles modified with P-CA (Ag@P-CA). The physical and chemical properties of P-CA and Ag@P-CA were compared with unmodified polyaniline (PANI) and PANI-modified Ag nanoparticles (Ag@PANI). The results of the characterization by X-ray photoelectron spectroscopy showed for the potential utility for the optimization of the adsorption/desorption and electron transfer steps during the electrochemical oxidation of nitrophenols. Ag@P-CA was used as an electrocatalyst by modifying polyaniline with Ag nanoparticles and citric acid, for the oxidation of 2-nitrophenol and 4-nitrophenol. These nanomaterials can significantly contribute to the identification of a strategy for detection and electrocatalytic remediation of nitrophenols and water.

The method of synthesizing these nanomaterials is original and can be used in various fields and industries. The authors have definitely proven a good knowledge and an important experience in the use of surface characterization methods.

I believe that the article could be improved if the authors take into account several aspects:

  1. The introduction is clear, short and to the point. Describe very well the properties of the nanomaterials of interest. However, it seems essential  to provide some examples from the specialized literature of synthesizing such modified nanoparticles used as electrocatalysts or other ways of detecting nitrophenols. 1-2 review paragraphs on this topic would improve the introduction.

Response:

The Introduction modified according to the discussion of the nitrophenols to address the reviewer comment.

  1. Figure 4 has two images A and B, but I don't see any difference and they are not mentioned in the text distinctly. Please verify or clarify.

Response:

Figure 4 was updated along with the accompanying text.

  1. In section 2.5., 1-2 citations would be useful to reinforce the interpretation of the parameters.

Response: Section 2.5 was updated accordingly.

  1. Why did you use CV as an electrochemical characterization method and not EIS, it is very suitable for evaluating the resistance to electron transfer.

Response: CV is one of the simplest electrochemical techniques which allows the chemist to quantitatively and qualitatively, not just study the electron transfer rate, but also to investigate the electrochemical and chemical reaction steps based on the basics of electrochemistry. Therefore, we believe that CV was the most suitable method for the current study. For gaining insight on the resistance of electron transfer, XPS characterization of the materials was also carried out.

  1. There are certain inappropriate expressions that appear in several places. For example "current density" line 341, 375 etc.

Response: Current density is referred to the measured current over the surface area of the working electrode to minimize the effect of surface area in our discussion.

  1. In section 2.8. CV would be useful to show the variation of the intensity of the oxidation current with the increase in the concentration of nitrophenols, possibly a calculation of the detection limit (if possible).

Response: Figure S6 wass added in the Supplementary Materials.

  1. In the conclusion section add some aspects regarding future research perspectives and possibilities of using these synthesized nanoparticles in several fields.

Response:

The conclusion was updated to address the reviewer comment.

  1. There are some phrases in the text that are too complicated and difficult to read. If possible, simplify the sentences to make it easier to read.

Response:  The manuscript was further edited for language, clarity, and syntax throughout to address the reviewer comment.

  1. Approximately half of the references are quite old. I recommend replacing some with newer references, at least from the last 10 years.

Response: 

We thank the reviewer for this comment. The citations were updated by comparing recent methods that employ various electrochemical and spectroscopic detection of nitrophenol, as summarized in Tables S1 and S2.

As well, the manuscript was further edited for language, clarity, and synthax throughout to meet the high standards of this journal.  We appreciate the constructive and insightful comments provided by Reviewer #1.

Reviewer 2 Report

The presented work (Manuscript ID, Title: Electrocatalytic oxidation of nitrophenols via Ag nanoparticles supported on modified citric acid-modified polyaniline, by the authors: Milad Khani, Ramaswami Sammynaiken, and Lee D. Wilson) has demonstrated the procedure for preparation of citric acid-modified polyaniline and unmodified polyaniline as support for Ag nanoparticles. The activity of nanocomposites was tested in the electrochemical oxidation of nitrophenols.  PANI-based composites were characterized with the aid of FTIR and NMR spectroscopy.  X-ray diffraction measurements reveal the crystalline structure and the size of Ag NPs deposited onto the composites. It was shown that Ag@P-CA has a higher conductivity in regard to Ag@PANI. Ag@P-CA was proposed as an electrocatalyst and a sensor material for nitrophenols. Several parameters that describe the sensor material are omitted.

Some considerations and questions are listed below:

§  What is presented in Fig. 4a and in Fig. 4b? It is not stated in the text, in the figure caption also.

§  What is the average size of Ag nanoparticles in composites according to Fig. 5b and Equation S1? It should be written in the text of the manuscript. From Fig. 5b it is not clear that the size of Ag NPs is the same using HCl or CA. Some explanation is necessary.

§  Did the activity depends on the size of Ag nanoparticles? Did the authors vary the content of Ag? What is the optimal content of Ag?

§  Concentration sensitivity was established but for a set of which concentrations?

§  Peak current density of Ag@P-CA increased linearly with increasing dye concentrations, which resulted in its sensitivity to the concentration of 2-NP and 4-NP in the solution. It is recommended to present a graph with the dependency of peak current density from concentration. What is the linear detection range? What is the value of sensitivity? Compare the obtained sensitivity of nitrophenols to various nanocomposites reported earlier.

§  Page 9, line 296: peak current density (ip), “peak” is missing

§  It is recommended to investigate the selectivity of sensor material for nitrophenols in the presence of organic compounds or cations.

§  In the experimental add if the new electrode was used for each dye concentration, or was the concentration continuously increased? Besides, was the first scan presented in Figure 9? It should be noted in the experimental part.

Author Response

Authors’ Response to Reviewer Comments on MS ID:  catalysts-2210212

Reviewer #2

The presented work (Manuscript ID, Title: Electrocatalytic oxidation of nitrophenols via Ag nanoparticles supported on modified citric acid-modified polyaniline, by the authors: Milad Khani, Ramaswami Sammynaiken, and Lee D. Wilson) has demonstrated the procedure for preparation of citric acid-modified polyaniline and unmodified polyaniline as support for Ag nanoparticles. The activity of nanocomposites was tested in the electrochemical oxidation of nitrophenols.  PANI-based composites were characterized with the aid of FTIR and NMR spectroscopy.  X-ray diffraction measurements reveal the crystalline structure and the size of Ag NPs deposited onto the composites. It was shown that Ag@P-CA has a higher conductivity in regard to Ag@PANI. Ag@P-CA was proposed as an electrocatalyst and a sensor material for nitrophenols. Several parameters that describe the sensor material are omitted.

Some considerations and questions are listed below:

  • What is presented in Fig. 4a and in Fig. 4b? It is not stated in the text, in the figure caption also.

Response:

Figure 4 is updated along with the accompanying text.

  • What is the average size of Ag nanoparticles in composites according to Fig. 5b and Equation S1? It should be written in the text of the manuscript. From Fig. 5b it is not clear that the size of Ag NPs is the same using HCl or CA. Some explanation is necessary.

Response:

A sentence was added to address the issue.

  • Did the activity depends on the size of Ag nanoparticles? Did the authors vary the content of Ag? What is the optimal content of Ag?

Response:  The size of Ag NPs are the same using these two acids, where an already established method was followed (Dolatkhah, A.; Jani, P.; Wilson, L.D. Redox-Responsive Polymer Template as an Advanced Multifunctional Catalyst Support for Silver Nanoparticles. Langmuir 2018, 34, 10560–10568, doi:10.1021/acs.langmuir.8b02336). In this study, an investigation was done to optimize the content of Ag in the composite materials.

  • Concentration sensitivity was established but for a set of which concentrations?

Response: The concentration range was added to the body of the article (section 2.8). Figure S6 is also added for more details.

  • Peak current density of Ag@P-CA increased linearly with increasing dye concentrations, which resulted in its sensitivity to the concentration of 2-NP and 4-NP in the solution. It is recommended to present a graph with the dependency of peak current density from concentration. What is the linear detection range? What is the value of sensitivity? Compare the obtained sensitivity of nitrophenols to various nanocomposites reported earlier.

Response:  Figure S6 (Supplementary Materials) is added for more details, and Table S2 is provided for a comparison of various methods and their sensitivity for detection of nitrophenols.

  • Page 9, line 296: peak current density (ip), “peak” is missing

Response:  The annotation was added as suggested by the reviewer.

  • It is recommended to investigate the selectivity of sensor material for nitrophenols in the presence of organic compounds or cations.

Response:  The focus of the article would be on the electrochemical oxidation of nitrophenols not detection of them. A study of the selectivity of the sensor material is outside of the original intended goal of this study. Instead, we have studied the selectivity between nitrophenol isomers.

  • In the experimental add if the new electrode was used for each dye concentration, or was the concentration continuously increased? Besides, was the first scan presented in Figure 9? It should be noted in the experimental part.

Response:  The information was added to the experimental part.

As well, the manuscript was further edited for language, clarity, and synthax throughout to meet the high standards of this journal.  We appreciate the constructive and insightful comments provided by Reviewer #2.

Round 2

Reviewer 2 Report

The authors responded to all questions and recognized all the suggestions. The manuscript can now be accepted.